Prepared for submission to JHEP

# A Gauge Theory for Shallow Water

**David Tong**

*Department of Applied Mathematics and Theoretical Physics*
*University Cambridge, CB3 0WA, UK*

*E-mail:* d.tong@damtp.cam.ac.uk

ABSTRACT: The shallow water equations describe the horizontal flow of a thin layer of fluid with varying height. We show that the equations can be rewritten as a $d = 2+1$ dimensional Abelian gauge theory. The magnetic field corresponds to the conserved height of the fluid, while the electric charge corresponds to the conserved vorticity. In a certain linearised approximation, the shallow water equations reduce to relativistic Maxwell-Chern-Simons theory. This describes Poincaré waves. The chiral edge modes of the theory are identified as coastal Kelvin waves.

## 1   Introduction

The shallow water equations describe the dynamics of a thin layer of fluid whose height is much smaller than its horizontal extent. The basic equations, and a number of extensions, are ubiquitous in modelling of the atmosphere, oceans, rivers, and lakes.

The variables of the shallow water equations are the height $h(x, y, t)$ of the fluid, and the horizontal velocity $\mathbf{u}(x, y, t)$. Writing the components of the 2d velocity vector as $u^i$, with $i = 1, 2$, the equations are

$$\frac{Dh}{Dt} = -h\nabla \cdot \mathbf{u} \quad \text{and} \quad \frac{Du^i}{Dt} = f\epsilon^{ij}u^j - g\frac{\partial h}{\partial x^i} \tag{1.1}$$

Here the material time derivative, $Dh/Dt = \partial h/\partial t + \mathbf{u} \cdot \nabla h$, tracks the variation of height with the flow. Similarly, $Du^i/Dt = \partial u^i/\partial t + (\mathbf{u} \cdot \nabla)u^i$.

The shallow water equations contain two parameters: the gravitational acceleration $g$, and the Coriolis parameter $f$. On Earth, at latitude $\theta$, the Coriolis parameter is given by $f = 4\pi \sin\theta \, (\text{day})^{-1}$. A derivation of the shallow water equations, together with many detailed applications can be found in the textbook [1]. A whirlwind tour of some highlights can be found in Section 4.3 of the lecture notes [2].

In recent years, a number of surprising parallels have been found between the shallow water equations, at least in their linearised form, and topological phases of matter. With a healthy dose of hindsight, the story goes back to the work of William Thomson who, in 1879, showed that the linearised shallow water equations admit chiral edge modes, now known as *coastal Kelvin waves* [3]. These waves are exponentially localised at the coast and propagate clockwise around land masses in the Northern hemisphere and anti-clockwise in the Southern hemisphere. Similar chiral modes in the shallow water equations are also found localised at the equator, now propagating from west to east [4, 5]. At a calculational level, these chiral waves are strikingly reminiscent of the chiral fermion zero modes of the Dirac equation that were discovered a century later [6].

In the quantum world, chiral edge modes are the smoking gun for a topological phase of matter [7–10]. This is also true for the classical shallow water equations, as first shown in the beautiful paper [11]. We will now give a cartoon version of the argument.

The linearised shallow water equations admit a band of solutions known as *Poincaré waves*, whose frequency $\omega$ is related to their wavevector $\mathbf{k}$ through the dispersion relation

$$\omega^2 = c^2 \mathbf{k}^2 + f^2 \tag{1.2}$$

Here the speed is given by $c = \sqrt{gH}$, with $H$ the average height of the fluid. (The derivation of this dispersion relation will be reviewed in Section 3.) For non-vanishing Coriolis force, $f \neq 0$, the frequency has a gap. But the gap closes at the equator where $f = 0$. Given such a gapped band of solutions, labelled by 2d momentum $\mathbf{k}$, one can follow lattice models and compute the Chern number by integrating a suitable Berry curvature over momentum space, á la TKNN [12] and Haldane [13]. This Chern number is non-vanishing, seemingly integer-valued, and depends on the sign of $f$. This means that the Chern class jumps as we cross the equator. This change in the underlying topology can be viewed as responsible for the existence of chiral equatorial waves.

The cartoon sketched above is sloppy for a number of reasons, not least the fact that momentum space in this problem is non-compact. This is in sharp contrast to the original uses of a Chern number which was for lattice models where momentum space is a compact Brillouin zone [12, 13]. This compactness is necessary for the Chern number to be an integer, and hence topological. To circumvent this issue, the original paper [11] used a slightly more subtle topological invariant, and subsequent papers have explored the possibility of regulating the momentum integral in some way [14, 15], although not without inducing further diversions [16]. Meanwhile, a topological explanation for the

original coastal Kelvin waves has also been presented, involving spatially varying $f(\mathbf{x})$ and mean height $H(\mathbf{x})$ [17]. These complications notwithstanding, the main lesson is clear: chiral modes observed in shallow water owe their existence to topology. Further studies of topological properties of the shallow water system can be found in [19–24].

The purpose of this work is to provide an effective field theory description of shallow water dynamics in the hope that it may make topological properties more manifest. We do this by writing the shallow water equations as a gauge theory. As we will see, this gauge theory involves Chern-Simons terms, which are the archetypal example of a topological field theory. In particular, chiral edge modes arise automatically in Chern-Simons theories, a fact that is familiar from quantum Hall physics [25, 26].

In Section 2, we show that the non-linear shallow water equations have a natural rendering as an Abelian gauge theory. The magnetic fields is associated to the height in the shallow water equations, while the electric charge is associated to the vorticity. We will see that the vorticity current must take a particular form, and therein lies many of the subtleties of the system.

In Section 3, we turn to the linearised equations. We will show that the effective field theory describing Poincaré waves (1.2) is given by relativistic Maxwell-Chern-Simons theory. The coastal Kelvin waves arise as chiral edge modes of this Chern-Simons theory.

## 2   A Gauge Theory for Shallow Water

In this section we formulate the shallow water equations as a gauge theory in $d = 2 + 1$ dimensions. Our strategy will be to identify conserved currents in the shallow water equations and identify them with appropriately conserved quantities in the gauge theory. Rather than simply write down the Lagrangian, we will instead proceed in smaller steps, and occasional missteps, to build some intuition for the gauge theoretic formulation. The more impatient reader can find the final answer in (2.9).

To understand the physics, we focus on conserved quantities. The shallow water equations (1.1) enjoy a number of conservation laws, but for our immediate purposes we need just two. Both are conserved quantities in the sense that they admit a current obeying the continuity equation,

$$\partial_\mu J^\mu = \frac{\partial J^0}{\partial t} + \nabla \cdot \boldsymbol{J} = 0$$

Throughout, we use $\mu = 0, 1, 2$ to denote space and time indices, and $i = 1, 2$ to denote only spatial indices. Note that, unlike in relativistic systems, $x^0 = t$ with no additional factor of speed. This means that $x^0$ and $x^i$ have different dimensions.

The first conserved quantity is the height $h$ of the fluid. Indeed, the first equation in (1.1) is a continuity equation with

$$J^0 = h \quad \text{and} \quad \boldsymbol{J} = h\mathbf{u} \tag{2.1}$$

This is a manifestation of the conservation of mass of the fluid. The second conserved quantity is associated to the vorticity, defined in two dimensions as

$$\zeta = \frac{\partial u^2}{\partial x} - \frac{\partial u^1}{\partial y}$$

The presence of the Coriolis force means that there is a slight modification to the conserved current, which is given by

$$\tilde{J}^0 = \zeta + f \quad \text{and} \quad \tilde{\boldsymbol{J}} = (\zeta + f)\mathbf{u} \tag{2.2}$$

Importantly, the two conserved currents (2.1) and (2.2) are not fully independent: this is because the 3-vectors $J^\mu$ and $\tilde{J}^\mu$ are always aligned,

$$\tilde{J}^\mu = \mathcal{Q}J^\mu \quad \text{with} \quad \mathcal{Q} = \frac{\zeta + f}{h} \tag{2.3}$$

The proportionality factor $\mathcal{Q}$ is known as the *potential vorticity*. From the expression (2.3), it follows that $\mathcal{Q}$ is materially conserved, meaning $D\mathcal{Q}/Dt = 0$. Part of the challenge will be to reproduce the dependent currents $J^\mu$ and $\tilde{J}^\mu$ from an action principle.

We now turn to the gauge theory. We denote our gauge field as $A_\mu$, with $\mu = 0, 1, 2$. In $d = 2 + 1$ dimensions, the electric field is a vector $E_i$, while the magnetic field $B$ is a (pseudo)-scalar,

$$E_i = \frac{\partial A_i}{\partial t} - \frac{\partial A_0}{\partial x^i} \quad \text{and} \quad B = \frac{\partial A_2}{\partial x^1} - \frac{\partial A_1}{\partial x^2}$$

Both are invariant under gauge transformations $A_\mu \to A_\mu + \partial_\mu \theta$.

Abelian gauge fields in $d = 2 + 1$ dimensions are special because they come with an associated conserved, global charge. This follows simply from the Bianchi identity which reads

$$\epsilon^{\mu\nu\rho}\partial_\mu\partial_\nu A_\rho = 0 \quad \Rightarrow \quad \frac{\partial B}{\partial t} - \frac{\partial E_2}{\partial x^1} + \frac{\partial E_1}{\partial x^2} = 0 \tag{2.4}$$

This has the form of the continuity equation, with the magnetic field as the charge and the components of the electric field as the current,

$$J^0 = B \quad \text{and} \quad \boldsymbol{J} = (-E_2, E_1) \quad \text{or} \quad J^i = -\epsilon^{ij}E_j$$

Currents that are conserved by virtue of the Bianchi identity are sometimes said to be *topological*.

To formulate a gauge theory description of shallow water, we will identify the topological current with the height current (2.1). This means that we take

$$B = h \quad \text{and} \quad E_i = \epsilon_{ij} h u^j \tag{2.5}$$

This single gauge field includes all degrees of freedom of the shallow water system, with the velocity given by $u^i = -\epsilon^{ij} E_j / B$. The first of the shallow water equations (1.1) is automatically obeyed due to the Bianchi identity (2.4). It remains to write the second shallow water equation in the language of the gauge theory. For this, we will adopt an action principle. We will make a first attempt at writing down an action for the shallow water system and see that it fails. But the way in which it fails will be instructive.

To motivate the action, we first note that the shallow water system has a conserved energy, given by

$$\mathcal{E} = \frac{1}{2} h \mathbf{u}^2 + \frac{1}{2} g h^2 = \frac{\mathbf{E}^2}{2B} + \frac{1}{2} g B^2$$

This takes the form of kinetic energy plus potential energy, so a natural guess for the action is

$$S_{\text{fail}} = \int dt \, d^2x \, \left( \frac{\mathbf{E}^2}{2B} - \frac{1}{2} g B^2 \right) \tag{2.6}$$

The fact that there is an inverse power of $B$ in the kinetic term should cause no more concern than an inverse power of the metric in the Einstein-Hilbert action: the dynamics should be thought of as an expansion around a non-vanishing fluid height, meaning that $B \neq 0$.

To derive the equations of motion, we vary the action with respect to $A_\mu$. The equation of motion for $A_0$ is Gauss' law and reads

$$\frac{\partial}{\partial x^i} \left( \frac{E^i}{B} \right) = 0 \quad \Rightarrow \quad \zeta = 0 \tag{2.7}$$

This is the first way in which the action (2.6) fails: Gauss' law only allows irrotational flows with the vorticity $\zeta$ forced to vanish. We'll rectify this shortly. For now, we note that that vorticity is the analog of electric charge in our theory. This restriction notwithstanding, we fare slightly better with the equations of motion for $A_i$. These are

$$\frac{\partial}{\partial t} \left( \frac{E_i}{B} \right) + \frac{1}{2} \epsilon_{ij} \frac{\partial}{\partial x^j} \left( \frac{\mathbf{E}^2}{B^2} \right) + g \epsilon_{ij} \frac{\partial B}{\partial x^j} = 0$$

When translated into the the variables $h$ and $\mathbf{u}$, the equation becomes

$$\dot{u}^i + \frac{1}{2} \frac{\partial \mathbf{u}^2}{\partial x^i} = -g \frac{\partial h}{\partial x^i} \quad \Rightarrow \quad \frac{Du^i}{Dt} + \epsilon^{ij} u^j \zeta = -g \frac{\partial h}{\partial x^i}$$

With the Gauss' law constraint $\zeta = 0$, this gives us something close to the shallow water equation (1.1). We see that, in addition to the fact the vorticity is forced to vanish, the action does not accommodate the Coriolis force. Of course, this should be no surprise as we made no attempt to include it.

With this failure behind us, we can now see how to resolve both issues. Clearly we need to include some additional degrees of freedom, charged under $A_\mu$, so that Gauss' law (2.7) receives an extra contribution, breathing life into the otherwise frozen vorticity. The question is: how can we introduce a vorticity current so that obeys the constraint (2.3)?

The way to achieve this is to introduce two, new scalar fields $\alpha$ and $\beta$, and couple them to the gauge field through the term

$$\tilde{J}^\mu = -\epsilon^{\mu\nu\rho} \, \partial_\nu \beta \, \partial_\rho \alpha \tag{2.8}$$

At the same time, we also introduce the Coriolis parameter $f$ through a background charge[1]. The resulting action is

$$S = \int dt \, d^2x \, \left( \frac{\boldsymbol{E}^2}{2B} - \frac{1}{2}gB^2 + fA_0 - \epsilon^{\mu\nu\rho} A_\mu \, \partial_\nu \beta \, \partial_\rho \alpha \right) \tag{2.9}$$

It is straightforward to see why the form of the current (2.8) gives the required result. If we vary the action with respect to the two scalar fields, $\beta$ and $\alpha$, we have the the the equations of motion

$$\epsilon^{\mu\nu\rho} F_{\nu\rho} \partial_\mu \beta = J^\mu \partial_\mu \beta = 0 \quad \text{and} \quad \epsilon^{\mu\nu\rho} F_{\nu\rho} \partial_\mu \alpha = J^\mu \partial_\mu \alpha = 0 \tag{2.10}$$

So both $\partial_\mu \alpha$ and $\partial_\mu \beta$ are orthogonal to the height current $J^\mu$. But, in 3d, this means that the cross-product of $\partial_\mu \alpha$ and $\partial_\mu \beta$ must be parallel to $J^\mu$. This cross-product is precisely the vorticity current $\tilde{J}^\mu$, defined in (2.8), and this reproduces the desired dependency (2.3).

We can rerun this same calculation in component form. First, Gauss' law for the revised action (2.9) is

$$\frac{\partial}{\partial x^i} \left( \frac{E_i}{B} \right) + f - \epsilon^{ij} \partial_i \beta \, \partial_j \alpha = 0 \quad \Rightarrow \quad \epsilon^{ij} \partial_i \beta \, \partial_j \alpha = \zeta + f \tag{2.11}$$

which is simply confirmation of the form of the current (2.8).

---

[1]As an aside: if the Coriolis parameter is time-dependent, so that $\dot{f} \neq 0$, then the action (2.9) is no longer gauge invariant. To remedy this, we should replace the term $fA_0$ with a coupling to a background current $f_\mu A^\mu$, with $f^0$ the Coriolis parameter and $\partial_\mu f^\mu = 0$. One can check that the additional terms $f^i A_i$ reproduce the Euler force.

Next, the equations of motion (2.10) read

$$B\dot{\alpha} + \epsilon^{ij} E_i \partial_j \alpha = 0 \quad \text{and} \quad B\dot{\beta} + \epsilon^{ij} E_i \partial_j \beta = 0 \tag{2.12}$$

From this we can construct an expression for the spatial part of the vorticity current (2.8), which is

$$\tilde{J}^i = \epsilon^{ij}(\dot{\beta}\partial_j \alpha - \dot{\alpha}\partial_j \beta) = -\frac{\epsilon^{ij} E_j}{B}\epsilon^{jk}\partial_k \alpha \partial_l \beta = -\frac{\epsilon^{ij} E_j}{B}(\zeta + f) = \frac{J^i}{J^0}\tilde{J}^0$$

This explicitly shows that the topological height current $J^\mu$ and the electric charge vorticity current $\tilde{J}^\mu$ lie parallel.

It remains only to compute the equation of motion for the original spatial gauge field, $A_i$. This is

$$\frac{\partial}{\partial t}\left(\frac{E_i}{B}\right) + \frac{1}{2}\epsilon_{ij}\frac{\partial}{\partial x^j}\left(\frac{\boldsymbol{E}^2}{B^2}\right) + g\epsilon_{ij}\frac{\partial B}{\partial x^j} - \epsilon_{ij}(\dot{\beta}\partial_j \alpha - (\partial_i\beta)\dot{\alpha}) = 0$$

This time, when translated into the the variables $h$ and $\mathbf{u}$, this equation coincides with the second shallow water equation (1.1). This confirms that the gauge theory action (2.9) does indeed describe the shallow water system. An alternative derivation of this action, using a duality transformation, is given in Appendix A.

The action (2.9) breaks time reversal invariance only when $f \neq 0$. In particular, the current intereaction $A \wedge d\beta \wedge d\alpha$ itself is invariant under time reversal which acts as $T : t \to -t$ and $\mathbf{x} \to \mathbf{x}$, together with

$$T : A_0 \to -A_0 , \quad A_i \to A_i , \quad \beta \to -\beta , \quad \alpha \to \alpha \tag{2.13}$$

Meanwhile, the action is invariant under parity which acts as $x^1 \to -x^1$ and $x^2 \to x^2$, together with

$$P : A_0 \to A_0 , \quad A_1 \to -A_i , \quad A_2 \to A_2 , \quad \beta \to \beta , \quad \alpha \to -\alpha \tag{2.14}$$

The $fA_0$ term also breaks charge conjugation, which acts as $A_\mu \to -A_\mu$.

Before we proceed, we note that the final term in (2.9) can be written as a Chern-Simons like form

$$\epsilon^{\mu\nu\rho} A_\mu \, \partial_\nu \beta \, \partial_\rho \alpha = \epsilon^{\mu\nu\rho} A_\mu \partial_\nu \tilde{A}_\rho \quad \text{with} \quad \tilde{A}_\mu = \partial_\mu \chi + \beta \partial_\mu \alpha \tag{2.15}$$

Any $d = 2+1$ dimensional gauge field can be written in the form above, a choice known as *Clebsch parameterisation*. This parameterisation is non-linear and non-unique. It does not often arise in quantum field theory but there is a long history of using this parameterisation in fluid mechanics, typically for the velocity field in three spatial dimensions. Here, we see that the shallow-water gauge theory (2.9) has what might be called a Clebsch-Chern-Simons term. We will later see that, in the linearised theory, this reduces to a genuine Chern-Simons term.

The fact that a Clebsch parameterisation of $\tilde{A}_\mu$ gives rise to currents $J^\mu$ and $\tilde{J}^\mu$ that are not fully independent is related to remarks make in [27, 28] about the role the Clebsch parameterisation plays in the symplectic structure of the Euler equation. Another approach to invoking Chern-Simons terms to describe rotating fluids was developed in [29], albeit with a rather different map between gauge fields and fluid variables. Related ideas, this time in the context of quantum Hall fluids, can also be found in [30].

## 3    The Linearised Shallow Water Equations

Much interesting physics, including aspects of topology, can be found in the linearised version of the shallow water equations. We expand about a fluid of constant, average height $H$, and write

$$h(\mathbf{x}, t) = H + \eta(\mathbf{x}, t)$$

Then, keeping only terms linear in $\eta$ and $\mathbf{u}$, the shallow water equations (1.1) become

$$\frac{\partial \eta}{\partial t} + H\nabla \cdot \mathbf{u} = 0 \quad \text{and} \quad \frac{\partial u^i}{\partial t} = f\epsilon^{ij}u^j - g\frac{\partial \eta}{\partial x^i} \tag{3.1}$$

In this section we first review some properties of the solutions to (3.1) and then turn to their description in the language of gauge theory.

### 3.1    Poincaré Waves

It is straightforward to find solutions to the linearised equations (3.1) by looking at Fourier modes. We write

$$\eta = \eta_0 e^{i(\omega t - \mathbf{k} \cdot \mathbf{x})} \quad \text{and} \quad \mathbf{u} = \mathbf{u}_0 e^{i(\omega t - \mathbf{k} \cdot \mathbf{x})}$$

The linearised shallow water equations (3.1) can then be written in a way that looks suggestively like the Dirac equation

$$\mathcal{H}\Psi = \omega\Psi \quad \text{where} \quad \mathcal{H} = \begin{pmatrix} 0 & ck_x & ck_y \\ ck_x & 0 & -if \\ ck_y & if & 0 \end{pmatrix} \quad \text{and} \quad \Psi = \begin{pmatrix} \sqrt{g/H}\eta_0 \\ u_0 \\ v_0 \end{pmatrix} \tag{3.2}$$

Here we've written the two-component velocity as $\mathbf{u}_0 = (u_0, v_0)$ and defined $c = \sqrt{gH}$ which we recognise as the speed of surface waves in the absence of the Coriolis force. We will see that $c$ plays a similar role in the present context.

The eigenvalue problem (3.2) is easily solved. There are two, gapped bands with frequencies

$$\omega = \pm\sqrt{c^2\mathbf{k}^2 + f^2} \tag{3.3}$$

These are known as *Poincaré waves*. For small wavelengths, $c|\mathbf{k}| \gg f$, these waves travel with speed $c$. The long-wavelength modes are affected by the presence of the Coriolis force.

In addition to the Poincaré waves, there is a flat band of solutions to (3.2), with

$$\omega = 0$$

This reflects the fact that there are static solutions to the full non-linear equations (1.1) beyond the obvious $h = \text{constant}$. These solutions have a non-trivial profile in one direction, say $h = h(y)$, with the gravitational force balanced by a corresponding velocity profile $fu = -\partial h/\partial y$ generating a Coriolis force. This is known as *geostrophic balance*. These static solutions manifest themselves in the linear problem as a flat band.

There is one other feature of the linearised shallow water equations (3.1) that we will need. This follows from conservation laws. The linearised version of the conservation of height (2.1) and vorticity (2.2) becomes

$$\frac{\partial \eta}{\partial t} + H\nabla \cdot \mathbf{u} = 0 \quad \text{and} \quad \frac{\partial \zeta}{\partial t} + f\nabla \cdot \mathbf{u} = 0$$

Eliminating $\nabla \cdot \mathbf{u}$ from both of these expressions gives us the surprisingly powerful formula

$$\dot{Q} = 0 \quad \text{with} \quad Q = H\zeta - f\eta \tag{3.4}$$

This is much stronger than most conservation laws: it is telling us that there is a function $Q(\mathbf{x})$ that does not change in time. This function $Q$ is, up to a rescaling[2], the linearised potential vorticity defined in (2.3).

We can ask: what is the potential vorticity evaluated on the solutions to the eigenvalue problem (3.2)? It is simple to check that the flat band, with $\omega = 0$, has

$$Q \sim \sqrt{\frac{H}{g}} \left(c^2\mathbf{k}^2 + f^2\right) \tag{3.5}$$

Meanwhile, the Poincaré waves with dispersion relation (3.3) have

$$Q = 0 \tag{3.6}$$

This fact will be important in the next section when we write down an effective field theory for the Poincaré waves.

---

[2]The non-linear potential vorticity is $\mathcal{Q} = (\zeta+f)/h$. Upon linearising, it becomes $\mathcal{Q} = fH + Q/H^2$.

## 3.2 A Linearised Gauge Theory

We now return to our gauge theory (2.9). We will see how the above phenomena arise in this language. We linearise the $A_\mu$ gauge fields as

$$A_\mu = \hat{A}_\mu + \delta A_\mu \quad \text{with} \quad \hat{A}_0 = 0 \quad \text{and} \quad \partial_1 \hat{A}_2 - \partial_2 \hat{A}_1 = H$$

We're going to abuse notation slightly and refer to the fluctuation $\delta A_\mu$ simply as $A_\mu$. (In other words, we make the substitution, $A_\mu \to \hat{A}_\mu + A_\mu$.) This will make the subsequent equations somewhat clearer. The translation (2.5) to the fluid variables then becomes

$$B = \eta \quad \text{and} \quad E_i = H \epsilon_{ij} u^j \tag{3.7}$$

which ensures that the first equation in (3.1) is obeyed courtesy of the Bianchi identity. Meanwhile, we also linearise the Clebsch gauge field $\tilde{A}_\mu$, with components

$$\beta = \hat{\beta} + p \quad \text{and} \quad \alpha = \hat{\alpha} + q \quad \text{with} \quad \partial_1 \hat{\beta} \, \partial_2 \hat{\alpha} - \partial_2 \hat{\beta} \, \partial_1 \hat{\alpha} = f \tag{3.8}$$

The fluctuations are written as $\delta\beta = p$ and $\delta\alpha = q$ to avoid an overload of $\partial\delta$'s in the expressions below. Expanding the action (2.9), we find that the terms linear in fluctuations vanish, as they must, while the terms quadratic in fluctuations read

$$S = \int dt \, d^2x \, \left( \frac{1}{2H} \boldsymbol{E}^2 - \frac{1}{2} g B^2 - H p \dot{q} + \epsilon^{ij} E_i \left( q \, \partial_j \hat{\beta} - p \, \partial_j \hat{\alpha} \right) \right) \tag{3.9}$$

The action is both simple and rather opaque. We will attempt to shed some light. First, we'll confirm that it reproduces the linearised shallow water equations. Then we'll look at the solutions. Gauss' law is

$$\partial_i E_i = H \epsilon^{ij} \left( \partial_i \hat{\beta} \, \partial_j q - \partial_i \hat{\alpha} \, \partial_j p \right) \tag{3.10}$$

while the equation of motion for $A_i$ is

$$\dot{E}_i = -g H \epsilon_{ij} \partial_j B - H \epsilon^{ij} \left( \partial_j \hat{\beta} \, \dot{q} - \partial_j \hat{\alpha} \, \dot{p} \right) \tag{3.11}$$

Finally, the equations of motion for $p$ and $q$ read

$$H \dot{q} = -\epsilon^{ij} E_i \partial_j \hat{\alpha} \quad \text{and} \quad H \dot{p} = -\epsilon^{ij} E_i \partial_j \hat{\beta} \tag{3.12}$$

We can substitute (3.12) into (3.11) to find

$$\dot{E}_i = \epsilon_{ij} \left( f E_j - g H \partial_j B \right) \tag{3.13}$$

This coincides with the second linearised shallow water equation in (3.1) using the dictionary (3.7).

As we've seen, the equations of motion have two classes of solution: the geostrophic flat band and the Poincaré waves. Our next goal is to disentangle these two solutions, and construct effective actions for each. This we now do in turn.

**An Effective Action for the Flat Band**

The geostrophic flat band consists time-independent solutions to (3.13) given by

$$A_0 = -\frac{c^2}{f}B$$

Given such a solution to (3.13), we should still self-consistently solve (3.10), (3.11) and (3.12) for the supplementary variables $q$ and $p$. In the present case, one can check that the solution is given by

$$Hq = -\epsilon^{ij}Z_i\partial_j\hat{\alpha} \quad \text{and} \quad Hp = -\epsilon^{ij}Z_i\partial_j\hat{\beta} \quad \text{with} \quad Z_i = \frac{c^2}{f}\left(\partial_i Bt - \frac{1}{f}\epsilon_{ij}\partial_j B\right)$$

These solutions obey the Gauss' law constraint (3.10) which takes the form

$$\partial_i E_i = \frac{c^2}{f}\nabla^2 B \tag{3.14}$$

An effective action for this flat band is given by

$$S = \int dt\, d^2x\, \frac{1}{2H}\left(E_i - \frac{c^2}{f}\partial_i B\right)^2 \tag{3.15}$$

It's straightforward to check that Gauss' law for this action coincides with (3.14). Meanwhile, the other equation of motion requires that $E_i - (c^2/f)\partial_i B$ is constant. Asymptotic conditions on the field require that this constant vanishes, reproducing the requirement for geostrophic balance.

**An Effective Action for Poincaré Waves**

Next, we turn to the Poincaré waves. Here, life is simpler if we work in the gauge $A_0 = 0$. We treat $p$ in (3.9) as a Lagrange multiplier, imposing the equation (3.12) for $q$. The linearised action (3.9) then becomes

$$S = \int dt\, d^2x\, \frac{1}{2H}\left(\dot{A}_i^2 - c^2 B^2 + f\epsilon^{ij}A_i\dot{A}_j\right) \tag{3.16}$$

where we have used the condition $\epsilon_{ij}\partial_i\hat{\beta}\,\partial_j\hat{\alpha} = f$. This should be accompanied by the Gauss' law constraint (3.10).

The Poincaré waves are solutions to the equations of motion arising from (3.16) of the form $A_i(\mathbf{x}, t) = e^{i(\omega t - \mathbf{k} \cdot \mathbf{x})} \bar{A}_i$, for some integration constants $\bar{A}_i$. One can check that the equations of motion impose the relativistic dispersion relation (3.3). The supplementary variables are now solved by

$$Hq = -\epsilon^{ij} A_i \partial_j \hat{\alpha} \quad \text{and} \quad Hp = -\epsilon^{ij} A_i \partial_j \hat{\beta}$$

with Gauss' law (3.10) now taking the form

$$\partial_i E_i = fB \tag{3.17}$$

Translating back to the fluid variables, this tells us that the potential vorticity is $Q = H\zeta - f\eta = 0$, in agreement with expectations from Poincaré waves in (3.6).

The upshot is that the Poincaré waves are described by the linearised action (3.9), together with the Gauss' law constraint (3.17). But these are the action and constraint of relativistic Maxwell-Chern-Simons theory. Indeed, we can reinstate $A_0$ in the action so that its equations of motion reproduce (3.17). The result is the familiar action

$$S = \int dt \, d^2x \, \frac{1}{2H} \left( \boldsymbol{E}^2 - c^2 B^2 - f \epsilon^{\mu\nu\rho} A_\mu \partial_\nu A_\rho \right) \tag{3.18}$$

This, of course, is Maxwell-Chern-Simons theory. Here, it governs the dynamics of Poincaré waves. It is simple to check that the profiles of the electric and magnetic fields coincide with the velocity and height of Poincaré waves, using the dictionary (3.7).

## 3.3 Coastal Kelvin Waves in Maxwell-Chern-Simons

Finally, we turn to edge modes in the shallow water system. The existence of such modes follows immediately from the description of the Poincaré waves as a Maxwell-Chern-Simons theory.

There is, of course, a great deal of discussion in the literature about the existence of edge modes in Chern-Simons theory, starting with the pioneering work of [31]. Indeed, these edge modes famously manifest themselves in the quantum Hall effect [25, 26]. Much of this discussion takes place in the quantum theory, where the idea of anomaly inflow plays a key role. But the quantum theory is not of much interest for our shallow water application. Nonetheless, as we now review, the existence of edge modes is not restricted to the quantum theory, and is a robust consequence of the classical dynamics.

The story is slightly different for pure Chern-Simons theories and for Maxwell-Chern-Simons theories. At long distances, the latter should be well approximated by the former so we start here. We consider the Chern-Simons action with a boundary at $x = 0$. Varying the action gives

$$S_{CS} = \int dt\, d^2x\; \epsilon^{\mu\nu\rho} A_\mu \partial_\nu A_\rho \quad \Rightarrow \quad \delta S_{CS} = \delta S_{\text{bulk}} + \int dt\, dy\; (A_2 \delta A_0 - A_0 \delta A_2)$$

where the bulk variation vanishes when the equations of motion are imposed (which, in this case, is just the condition $F_{\mu\nu} = 0$.) To have a well defined variational principle, we should pick a boundary condition so that the second term above also vanishes. There are two simple choices: we could pick either $A_0 = $ constant or $A_2 = $ constant.

In the presence of a boundary, Chern-Simons theory comes with an additional concern. The Chern-Simons action $S_{CS}$ is invariant under gauge transformations only up to a total derivative. This means that the theory risks a failure of gauge invariance in the presence of a boundary. Under a gauge transformation $A_\mu \to A_\mu + \partial_\mu \theta$, the action (3.18) transforms as

$$S_{CS} \to S_{CS} + \int dt\, dy\; \theta\, E_2 \tag{3.19}$$

For pure Chern-Simons theory, neither of the boundary conditions is sufficient to set $E_2 = 0$ on the boundary, so we're in trouble. The resolution is that one should restrict to gauge transformations $\theta$ that vanish on the boundary. But restricting the allowed gauge transformations resurrects modes that were previously gauge redundancies [31]. These are the edge modes. A detailed classical analysis of these modes can be performed by carefully looking at the Poisson bracket structure in the presence of a boundary [33].

Note that pure Chern-Simons theory has no parameter that can play the role of a speed. This means that, without additional information, these edge modes have no dynamics. This additional information could be provided by a boundary Hamiltonian, or by some deformation of the bulk theory. (We'll see an example of the latter below.) Only after this perturbation do the edge modes reveal their chiral nature [25, 26, 32].

Next, we turn to Maxwell-Chern-Simons theory. This, as we've seen, is the effective description of Poincaré waves. Varying the action (3.18), again with a boundary at $x = 0$, gives

$$\delta S = \delta S_{\text{bulk}} + \frac{1}{2H} \int dt\, dy\; \left[ -(2E_1 - fA_2)\delta A_0 - (2c^2 B + f A_0)\delta A_2 \right] \tag{3.20}$$

Again, the bulk term $\delta S_{\text{bulk}}$ vanishes when the equations of motion are obeyed. Again, we need to think about what boundary conditions we can impose. Because the Maxwell-Chern-Simons action has two derivatives, rather than just one, the phase space has twice

the dimension of pure Chern-Simons theory. This means that we get to impose two boundary conditions rather than just one. A particularly natural boundary condition is $A_0 = $ constant *and* $A_2 = $ constant. Note that the choice of constant splits the theory up into superselection sectors.

Importantly, any choice of constant ensures that $E_2 = 0$ on the boundary. This means that the gauge variation of the Chern-Simons action (3.19) vanishes. However, we must still restrict to gauge transformations that do not change our choice of $A_0$ and $A_2$ on the boundary. The Poisson bracket analysis for Maxwell-Chern-Simons theory was performed in [34] (albeit in AdS space, rather than flat space but the essential details for our purposes are unchanged.)

In electromagnetism, the boundary condition $E_2 = 0$ is appropriate for a conductor. In our shallow water system, it is the boundary condition of choice because it translates to the requirement that $\mathbf{u} \cdot \hat{\mathbf{x}} = 0$, so that no fluid flows into the boundary.

What becomes of the Chern-Simons edge modes now that we have a Maxwell term in the bulk? From our previous discussion, we would expect the perturbation to breathe life into these modes. Indeed, this happens in a very appealing way. The bulk equations of motion of Maxwell-Chern-Simons theory are

$$\partial_i E_i = fB \quad \text{and} \quad \dot{E}_i = -c^2 \epsilon_{ij} \partial_j B + f \epsilon_{ij} E_j$$

The boundary condition is $A_0 = A_2 = 0$. We can look for solutions that obey $A_0 = A_2 = 0$ throughout the bulk. We make the ansatz

$$A_1 = A(x) \, e^{i(\omega t - ky)}$$

The bulk equations tell us that

$$\dot{E}_1 = -c^2 \partial_2 B \quad \Rightarrow \quad \omega^2 = c^2 k^2 \quad \Rightarrow \quad \omega = \pm ck$$

and

$$\partial_1 E_1 = fB \quad \Rightarrow \quad \omega A' = kfA \quad \Rightarrow \quad A' = \pm \frac{f}{c} A$$

For $f > 0$, corresponding to the Northern hemisphere, a fluid restricted to lie in the region $x > 0$ has the normalisable solution only if

$$\omega = -ck \quad \Rightarrow \quad A(x) \sim e^{-fx/c}$$

This is the chiral, coastal Kelvin wave, first found in [3]. It is the manifestation of the Chern-Simons edge modes, now appearing as a classical solution of Maxwell-Chern-Simons theory. The Kelvin wave travels in the direction of decreasing $y$ which, in this context, means southward. Said differently, the wave moves around land masses in the clockwise direction in the Northern hemisphere. The novelty here is to see this wave emerging from the effective Maxwell-Chern-Simons description.

The fact that the edge modes of Chern-Simons theory become classical solutions of Maxwell-Chern-Simons theory was, to my knowledge, first noted in [35], where the authors study the connection to anomaly inflow.

# 4 Discussion

We have shown the shallow water equations have a gauge theoretic description. In the linearised theory, solutions neatly split into the geostrophic flat band and Poincaré waves. The flat band is described by the gauge theory (3.15), while Poincaré waves are described by Maxwell-Chern-Simons theory (3.18). The well-known edge modes of Chern-Simons theory are then identified as coastal Kelvin waves.

There are a number of interesting open questions. First, the properties of coastal Kelvin waves were studied in the presence of Hall viscosity in [16] and certain anomalous behaviour was encountered, with the number of edge modes depending on the choice of boundary conditions. Does the same behaviour arise in the Chern-Simons description, presumably after implementing the Hall viscosity in some way? If so, does this, in turn, have implications for quantum Hall systems?

Second, we have restricted ourselves here to the situation in which the Coriolis parameter, $f$, is constant. If $f$ depends on space (but not on time) then the non-linear action (2.9) remains valid, as too does the linearised theory (3.9). It remains, however, to disentangle the two branches of solutions and construct effective theories for each. In particular, the geostrophic flat band famously picks up a dispersion when $\nabla f \neq 0$, giving rise to Rossby waves. One interesting challenge is to extend the flat band effective theory (3.15), to derive a gauge theoretic formulation of the so-called quasi-geostrophic equation that governs the dynamics of Rossby waves.

Relatedly, finding the appropriate generalisation to situations with $\nabla f \neq 0$ may give insight into the topological nature of equatorial waves. This, of course, was the focus of the original paper revealing topological structure in shallow water [11]. There, the topology of the Poincaré band implied the existence of two chiral, equatorial modes and these were identified with the Kelvin wave and Yanai wave. The Kelvin wave is uncomplicated: like the Poincaré waves it has potential vorticity $Q = 0$, which means that it doesn't mix with the (now almost) flat band of Rossby waves. But the Yanai wave is more mysterious since, in condensed matter language, it hybridises with the Rossby waves. In particular, it has non-vanishing potential vorticity, and it is not clear how such a wave can be constructed from bulk modes that have strictly $Q = 0$. Perhaps these features of the Yanai wave go beyond topological considerations, but they are qualitatively striking and it would be nice to find some deeper explanation of these properties.

Finally, and more generally, gauge theories have long proven themselves to be an arena in which subtle and surprising topological effects arise. Hopefully it will be a useful framework to find further effects in fluid dynamics.

## A   An Alternative Derivation of the Gauge Theory

There is a long history of constructing variational principles for fluids in the Eulerian framework. The results do not usually look like a gauge theory. In this appendix we provide an alternative derivation of the gauge theory action (2.9) that makes contact with the more traditional approach.

For fluids in $d$ spatial dimensions, one usually starts by introducing a map from physical spacetime to the positions of fluid parcels.

$$\alpha : \mathbb{R}^d \times \mathbb{R} \to \mathbb{R}^d$$

We write this map as $\alpha^a(\mathbf{x}, t)$ with $a = 1, \ldots, d$. The map must be a diffeomorphism, with $\det(\partial \alpha^a / \partial x^i) \neq 0$. A discussion of this approach to fluids can be found in the textbook [36]. More recently, it has been revived in the context of relativistic fluids [37, 38].

Given a map $\alpha^a(\mathbf{x}, t)$, the velocity field $\mathbf{u}$ is defined by the equation

$$\frac{D\alpha^a}{Dt} \equiv \frac{\partial \alpha^a}{\partial t} + \mathbf{u} \cdot \nabla \alpha^a = 0 \tag{A.1}$$

This captures the idea that the physical label $\alpha^a(\mathbf{x}, t)$ of the point in the fluid is unchanged as the fluid moves. The velocity $\mathbf{u}$ defined in this way is invariant under diffeomorphisms of $\alpha$. Note, however, that this definition comes with a rather confusing minus sign, related to active vs passive transformations. This is seen, for example, in the simplest laminar flow $\boldsymbol{\alpha} = (x - ut, y)$ which, according to (A.1), is associated to a velocity field $\mathbf{u} = (+u, 0)$.

For application to the shallow water equations, we have $d = 2$ dimensions. We also have the height variable $h(x, y, t)$, which obeys the continuity equation

$$\frac{Dh}{Dt} + h\nabla \cdot \mathbf{u} = 0 \tag{A.2}$$

reflecting the incompressibility of the underlying fluid.

An action for the shallow water equations in these variables was given in [39]

$$S = \int dt\, d^2x \left( \frac{1}{2}h\mathbf{u}^2 - \frac{1}{2}gh^2 - h\mathbf{u}\cdot\mathbf{a}(\mathbf{x}) \right) + \phi \left( \frac{Dh}{Dt} + h\nabla\cdot\mathbf{u} \right) - h\beta_a \frac{D\alpha^a}{Dt}$$

This action should be varied with respect to $h$, $\mathbf{u}$, $\phi$, $\alpha^a$ and $\beta_a$. Both $\beta_a$ and $\phi$ act as Lagrange multipliers, imposing the conditions (A.1) and (A.2) respectively. We have introduced a background gauge field $\mathbf{a}(\mathbf{x})$ whose role will be to implement the Coriolis force[3]. This background field depends only on $\mathbf{x}$ and not on time. We will determine the form of $\mathbf{a}(\mathbf{x})$ shortly.

The equation of motion for $\mathbf{u}$ reveals that the velocity has a Clebsch-like parameterisation,

$$\mathbf{u} = \nabla\phi + \beta_a \nabla\alpha^a + \mathbf{a}(\mathbf{x}) \tag{A.3}$$

This differs from our previous Clebsch parameterisation for the gauge field (2.15) in that we have a pair of potentials $(\beta_a, \alpha_a)$, with $a = 1, 2$, with $\alpha^a$ restricted to be a diffeomorphism. In addition, the velocity gets a contribution from the fixed background $\mathbf{a}(\mathbf{x})$.

The equation of motion for $\alpha^a$ reveals a conserved current, involving the Lagrange multiplier $\beta_a$,

$$\frac{\partial(h\beta_a)}{\partial t} + \nabla\cdot(h\beta_a\mathbf{u}) = 0 \quad \Rightarrow \quad \frac{D\beta_a}{Dt} = 0 \tag{A.4}$$

where the second equation follows from the conservation of the height (A.2). Finally, the equation of motion for $h$ gives

$$\frac{D\phi}{Dt} = \frac{1}{2}\mathbf{u}^2 - gh - \mathbf{u}\cdot\mathbf{a} \tag{A.5}$$

where an additional term has been set to zero courtesy of (A.1). Although it is not immediately obvious, equations (A.1), (A.3), (A.4) and (A.5) are equivalent to the second of the shallow water equations (1.1). To see this, we first use (A.3) to compute the material derivative of the velocity

$$\frac{Du_i}{Dt} = (\partial_i\dot\phi + u^j\partial_j\partial_i\phi) + (\dot\beta_a + u^j\partial_j\beta_a)\partial_i\alpha^a + \beta_a(\partial_i\dot\alpha^a + u^j\partial_j\partial_i\alpha^a) + u^j\partial_j a_i$$

---

[3]In quantum Hall physics, there is a convention that lower-case gauge fields $a_\mu$ are dynamical while upper-case gauge fields $A_\mu$ are background. Annoyingly, we have chosen the opposite convention here. As we will see, this has the advantage that the field strength $f_{12} = \partial_1 a_2 - \partial_2 a_1$ coincides with the Coriolis parameter, canonically denoted in geophysics as $f$.

Invoking (A.1), (A.4) and (A.5), a short calculation then reveals

$$\frac{Du_i}{Dt} = -f_{ij}u^j - g\partial_i h \quad \text{with } f_{ij} = \partial_i a_j - \partial_j a_i$$

We see that the background field strength $f_{ij}$ acts like a Coriolis force. To reproduce the constant Coriolis force of (1.1), we simply need to find a background $\mathbf{a}(\mathbf{x})$ that satisfies $f_{ij} = -f\epsilon_{ij}$. For example, $\mathbf{a} = f(y,0)$ or $\mathbf{a} = \frac{1}{2}f(y,-x)$ both do the job. Note that any choice necessarily breaks translational symmetry and/or rotational symmetry, even though the final shallow water equations do not. This is unavoidable in this formalism. The situation is identical to that of a charged particle in a constant background magnetic field.

We now show how this formalism is related to the gauge theory introduced in Section 2. The idea is to use a 3d duality transformation, first introduced in [40] and beloved of many a high-energy and condensed matter theorist. To proceed, it's useful to first integrate by parts, so that the Lagrangian has an overall factor of the height $h$. (This reflects the fact that the shallow water equations arise after integrating over the $z$-direction of the thin fluid.)

$$S = \int dt\, d^2x\; h \left(\frac{1}{2}\mathbf{u}^2 - \frac{1}{2}gh - \mathbf{u}\cdot\mathbf{a}(\mathbf{x}) - \frac{D\phi}{Dt} - \beta_a\frac{D\alpha^a}{Dt}\right) \tag{A.6}$$

Note that only derivatives of the Lagrange multiplier $\phi$ appear in this action. This allows us to exchange $\phi$ for a vector field $C_\mu$, by writing the action as

$$S = \int dt\, d^2x\; h \left(\frac{1}{2}\mathbf{u}^2 - \frac{1}{2}gh - \mathbf{u}\cdot\mathbf{a}(\mathbf{x}) - C_0 - u^i C_i - \beta_a\frac{D\alpha^a}{Dt}\right) + \epsilon^{\mu\nu\rho}A_\mu\partial_\nu C_\rho$$

Here we have further introduced a new Lagrange multiplier, $A_\mu$. This can be viewed as a gauge field because the action is invariant under transformation $A_\mu \to A_\mu + \partial_\mu\theta$. The equation of motion for $A_\mu$ gives the constraint $\epsilon^{\mu\nu\rho}\partial_\nu C_\rho = 0$ which can be solved locally by $C_\mu = \partial_\mu\phi$. This takes us back to our original action (A.6). Alternatively, we may instead choose to impose the equation of motion for $C_\mu$, which also acts as a Lagrange multiplier. This gives a constraint on the magnetic field $B = \partial_1 A_2 - \partial_2 A_1$ and electric fields $E_i = \dot{A}_i - \partial_i A_0$,

$$B = h \quad \text{and} \quad E_i = \epsilon_{ij}hu^j$$

These coincide with the relations introduced in (2.5). We may then replace $h$ and $\mathbf{u}$ in the action with our new gauge fields. This gives

$$S = \int dt\, d^2x\; \left(\frac{\mathbf{E}^2}{2B} - \frac{1}{2}gB^2 + \epsilon^{ij}E_i a_j(\mathbf{x}) - \epsilon^{\mu\nu\rho}\partial_\mu A_\nu\,\beta_a\partial_\rho\alpha^a\right)$$

In its essential details, this coincides with the action (2.9) presented in Section 2. The Coriolis term becomes the background charge $fA_0$ after an integration by parts. The form above makes it clear that this term too has its origin as a Chern-Simons term, now involving a background field. After integration by parts, the action respects both translation and rotation symmetry provided that $f$ is constant. Meanwhile, the second gauge field arises automatically in the Clebsch parameterisation in the above presentation,

$$\tilde{A}_\mu = \partial_\mu \chi + \beta_a \partial_\mu \alpha^a$$

Here $\chi$ is pure gauge and does not contribute to the action. This differs from (2.15) only in the $a = 1, 2$ index ascribed to $\alpha$ and $\beta$. Diffeomorphism invariance ensures that this does not contribute any further degrees of freedom.

There are other equations in fluid mechanics that have a similar structure to the shallow water equations (1.1) and hence lend themselves to a gauge theoretic formalism. For example, a compressible fluid obeys the equations

$$\frac{\partial \rho}{\partial t} + \nabla \cdot (\rho \mathbf{u}) = 0 \quad \text{and} \quad \rho \frac{D\mathbf{u}}{Dt} = -\nabla P$$

with an equation of state relating the pressure $P$ to the density $\rho$ of the form $P = C\rho^\gamma$, where $C$ is a constant and $\gamma$ is the ratio of specific heats. When the fluid lives in $d = 2$ spatial dimensions, this too may be written as gauge theory, with $B = \rho$ and $E_i = \epsilon_{ij} \rho u^j$ and with the potential term $B^2$ in (2.9) replaced by $B^{\gamma-1}$. Again, the slightly disconcerting fact that the power $\gamma$ is not an integer is no cause for concern when it is appreciated that this theory should be expanded around the solution $B = \rho = \text{constant} \neq 0$.

### Acknowledgements

I'm especially grateful to Matt Davison for introducing me to the idea of topology in fluid dynamics and for guiding me through some basic aspects of atmospheric and oceanic flows. Thanks also to Jay Armas, Jackson Fliss, Sriram Ganeshan, Sean Hartnoll, Max Metlitski, Gustavo Monteiro, V. Parameswaran Nair, and David Skinner for helpful discussions. This work was supported by STFC grant ST/L000385/1, the EPSRC grant EP/V047655/1 "Chiral Gauge Theories: From Strong Coupling to the Standard Model", and a Simons Investigator Award. For the purpose of open access, the author has applied a Creative Commons Attribution (CC BY) licence to any Author Accepted Manuscript version arising from this submission.

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
