# Peer review of "A Gauge Theory for Shallow Water"

_SciPost Physics_

## Round 2 · Referee Report · Brad Marston (Referee 1) · 2022-11-29

Strengths

  1. The paper provides a complementary perspective on fluid waves of topological origin by formulating the fluid dynamics as a gauge theory with an added Chern-Simons term.

  2. The manuscript is well-written and of significant pedagogical value.

  3. The gauge formulation may be able to tackle new classes of problems.

  4. An Appendix makes contact with the more standard variational approach to fluid dynamics that will invite readers to think more about variational approaches to fluid dynamics.

Weaknesses

  1. The role of discrete symmetries, in particular the breaking of time-reversal and reflection symmetries, should be discussed.

  2. The paper ends rather abruptly and would be significantly strengthened by the addition of a Discussion or Conclusion section at the end that would reflect on the import and limitations of the gauge formulation. The following questions come to mind: (a) Does the gauge theory circumvent the problem with computing Chern numbers in continuous fluid systems (non-compactness in wavevector space)? If so, how? (b) Could it permit insight into non-Hermitian physics such as driving and dissipation? (c) Is there a path forward for the incorporation of nonlinearity into the topological theory? (d) Generally what is the significant or importance of the new formulation?

  3. It would be helpful if all equations that are set on their own lines were numbered.

  4. I don't understand the sentences on page 11: "While (3.13) coincides with the action for Maxwell-Chern-Simons theory in the A0 = 0 gauge, this is not the Gauss' law for that theory. That's why these flat band solutions are unfamiliar in the context of Chern-Simons theory." I understand that the flat bands don't arise in the usual Maxwell-Chern-Simons theory but I am not following the reasoning here.

  5. The manuscript discusses the topological origin of coastal Kelvin waves but not the equatorial Kelvin and Yanai waves. It would be useful for the reader if a section regarding that were added (or if there is some problem with the understanding the equatorial waves, that should be noted).

Report

The manuscript meets 2 of the 4 expectations for publication in SciPost (only 1 is required). It "opens a new pathway in an existing or a new research direction, with clear potential for multipronged follow-up work" and it "provides a novel and synergetic link between different research areas."

The writing is exceptionally clear and I will be able to recommend publication once the weaknesses that are identified above are addressed.

Requested changes

Please address the points raised above.

  • validity: high
  • significance: good
  • originality: good
  • clarity: top
  • formatting: excellent
  • grammar: perfect

Author:  David Tong  on 2022-12-23  [id 3185]

(in reply to Report 1 by Brad Marston on 2022-11-29)

Dear Professor Marston,

Thank you for your detailed reading and comments. In reply to your points:

-- Discrete symmetries: Yes. There's nothing very much new to say as everything carries over in a straightforward way to the gauge theory. I've added the relevant equations in (2.13) and (2.14)

-- I added a discussion section. The questions that you bring up are all interesting. I mentioned only a subset of them.

-- Sorry to be unclear on page 11. I rewrote the whole section so that it's hopefully more understandable. The essence is that the non-linear theory captures both the flat band and the Poincare waves and the goal is to disentangle them, to get effective theories for each. I tried to do it in one foul swoop before but it's clearer to do them one at a time, not least because the story is simplest if we pick slightly different gauge fixing conditions for each.

-- I don't yet understand how to see the topological origin of equatorial waves in the gauge theory framework. That's on the to-do list.

Best Wishes, David

---

## Round 2 · Referee Report · Anonymous (Referee 2) · 2022-12-1

Report

In this paper David Tong formulates a gauge theory that gives rise to hydrodynamic shallow water equations under external rotation. The gauge formulation is subtle because the two conserved currents in this problem are not independent. The author derives the Poincare and coastal Kelvin waves. This effective theory approach is satisfying because it equips us with a simple-looking action. Moreover, it is useful because in the future one can include and investigate systematically higher-order corrections to the standard shallow water equations.

I would like the author to address the following comments:

1) In the abstract it is claimed that the theory contains two gauge fields. However, due to the interdependence of the two currents, it appears that we have only one $U(1)$ gauge redundancy with a gauge field $A_\mu$ and two (?gauge-invariant?) scalars $\alpha$ and $\beta$. If this is correct, why does the author prefer to talk about the gauge field $\tilde A_\mu$ with its additional independent $U(1)$ gauge redundancy?

2) Imagine the Coriolis parameter is time-dependent and the action (2.7) is not gauge-invariant anymore. How could this be remedied?

3) In the calculation of the coastal Kelvin waves on page 13, given that there is only one physical boundary condition $\mathbf{u}\cdot \hat {\mathbf{x}}=0$, why two boundary condition $A_0=A_2=0$ are actually prescribed? Relatedly, why are the above two conditions later used in the bulk instead of only fixing them on the boundary?

I think the paper is useful and should be published in SciPost Physics after the above comments are addressed.

  • validity: high
  • significance: high
  • originality: high
  • clarity: top
  • formatting: perfect
  • grammar: perfect

Author:  David Tong  on 2022-12-23  [id 3186]

(in reply to Report 2 on 2022-12-01)

Thank you for the detailed reading of the paper and the comments. Here are some replies to your points:

-- I originally formulated the action with two gauge fields, one the a familiar gauge field A_\mu and the other a Clebsch-parameterised gauge field \tilde{A}_\mu. This was partly to make contact with earlier work on the Poisson structure of fluid equations. But, upon reflection, I think it's best to write things just in terms of the "Clebsch parameters" \alpha and \beta. I've restructured the derivation in Section 2 to reflect this, and relegated the observation that things can be written in terms of \tilde{A} to a small comment at the end of Section 2.

-- What happens if f is time dependent? This is a great question and the answer brought a small smile to my face. First, gauge invariance means that you need to change the background charge $f A_0$ to a full current $f_\mu A^\mu$, with $\partial_\mu f^\mu =0$. This means that if f is time dependent, you'll necessarily get extra terms $f_i A^i$ in the action. If you work out what these extra terms are then (assuming suitable symmetries), you find that they capture the Euler force. (This is the force proportional to $\dot{\omega}$ that I always struggle to illustrate with interesting examples when teaching it to undergraduates.) It strikes me as quite cute that this is related to the Coriolis force by gauge invariance in this framework. I added a small footnote (omitting the details) on page 6 explaining this.

-- On the boundary condition: $E_2=0 $on its own is not sufficient to make the boundary term vanish. You need both $A_0 = constant$ and $A_2=constant$. I clarified this in the paper.

When looking for the bulk solution, you're not obliged to put $A_0=A_2=0$ everywhere of course. But if you make that ansatz then you find the Kelvin wave.

Thanks again for your comments. Best Wishes and Happy Holidays
David

---

## Editorial Decision

resubmitted